# High-Resolution Magnetoelectric Sensor and Low-Frequency Measurement Using Frequency Up-Conversion Technique

**DOI:** 10.3390/s23031702

**Published:** 2023-02-03

**Authors:** Kunyu Sun, Zhihao Jiang, Chengmeng Wang, Dongxuan Han, Zhao Yao, Weihua Zong, Zhejun Jin, Shandong Li

**Affiliations:** 1College of Physics, Center for Marine Observation and Communication, Qingdao University, Qingdao 266071, China; 2College of Electronics and Information, Qingdao University, Qingdao 266071, China

**Keywords:** magnetoelectric sensors, frequency up-conversion technique, piezomagnetic material, low-frequency weak magnetic field

## Abstract

The magnetoelectric (ME) sensor is a new type of magnetic sensor with ultrahigh sensitivity that suitable for the measurement of low-frequency weak magnetic fields. In this study, a metglas/PZT-5B ME sensor with mechanical resonance frequency $f_{res}$ of 60.041 kHz was prepared. It is interesting to note that its magnetic field resolution reached 0.20 nT at $f_{res}$ and 0.34 nT under a DC field, respectively. In order to measure ultralow-frequency AC magnetic fields, a frequency up-conversion technique was employed. Using this technique, a limit of detection (LOD) under an AC magnetic field lower than 1 nT at 8 Hz was obtained, and the minimum LOD of 0.51 nT was achieved at 20 Hz. The high-resolution ME sensor at the sub-nT level is promising in the field of low-frequency weak magnetic field measurement technology.

## 1. Introduction

With the development of the information age, the Internet of Things requires all kinds of sensors to accurately detect the physical properties of different substances, leading to wide applications of sensors. For example, microelectromechanical system (MEMS) sensors can work in a variety of environments [1,2,3,4,5,6]. The magnetic sensor is one kind of the most important sensors. According to the working principle, magnetic sensors can be divided into the following types: magnetoresistance types (for example, anisotropic magnetoresistance, giant magnetoresistance, and tunneling magnetoresistance), magnetic flux types (fluxgate magnetometer, optic-pumping atomic magnetometer, proton precession magnetometer, and superconducting quantum interference devices), magnetic field types (giant magnetoimpedance and magnetoelectric coupling sensor), and so on. Most magnetoresistance sensors have relatively low magnetic field sensitivity; however, recently, the sensitivity of tunneling magnetoresistance sensors has been greatly improved, in the order of sub-pT [7]. Magnetic flux-type sensors exhibit very high magnetic field resolution, but the magnetic field resolution strongly depends on the dimension of the sensor. The reduction in dimensions results in a dramatic deterioration of magnetic field resolution [8,9,10,11]. Although the performance of the ME sensor also depends on its size, its size dependence is not as strong as that of the flux-type sensor, which is beneficial to its miniaturization [12,13,14]. At present, ME sensors have been used for non-contact positioning and navigation [15,16,17,18,19,20,21]. With the continuous progress of technology, ME sensors are expected to be applied in the geomagnetic field and magnetoencephalography in the near future.

ME sensors, which are based on the magnetoelectric coupling effect, can timely transmit magnetic field signals as voltage, mediated by interfacial stress (σ) [3,22,23,24,25]. A good ME sensor should have a high ME coupling coefficient ($\alpha_{ME}$); thus, large output voltage is available under a certain external magnetic field. The ME conversion efficiency reaches a maximum when the frequency of the external AC field is close to the mechanical resonance frequency ($f_{res}$) of the sensor. A DC bias magnetic field $(H_{bias}$) is usually exerted on the piezomagnetic layer to maximize the piezomagnetic coefficient ($d_{m}$). Therefore, the best magnetic field sensitivity of an ME sensor with optimized $\alpha_{ME}$ should be working at $f_{res}$ under an optimal DC bias magnetic field (in active modes, the bias field may be as small as zero) [11,26,27,28,29]. Recently, research work has focused on reducing $f_{res}$ and improving $\alpha_{ME}$ in ME sensors. D. Viehland et al. reported self-biased Metglas/Pb(Zr,Ti)O_3_/Metglas laminates with $\alpha_{ME}{}_{\;}=12\;V/\mathrm{cm}\cdot \;\mathrm{Oe}$ and 380 V/cm⋅ Oe at 1 kHz and electromechanical resonance [26]. E. Quandt et al. enhanced the magnetoelectric coupling coefficient up to ${\left|\alpha_{ME}\right|}_{max}=6.9\;\mathrm{kV}/\mathrm{cm}\cdot \;\mathrm{Oe}@H_{bias}=2.4\;\mathrm{Oe}$ in an AlN/FeCoSiB thin-film sensor and reduced $f_{res}$ to around 870 Hz [30].

In general, the frequency of the weak magnetic field to be measured is usually lower than 1 kHz (such as magnetoencephalography, magnetocardiography, and underwater/underground mineral resources), while the typical operating frequency of bulk ME sensors is several tens of kHz. Moreover, the magnetic field resolution dramatically reduces with deviation from $f_{res}$. In order to sensitively measure ultra-low-frequency magnetic field signals using an ME sensor vibrating at $f_{res}$, a frequency up-conversion technique was employed in this study. If the low-frequency ($f_{tbm}$) magnetic field to be measured is exerted on the ME sensor vibrating at its $f_{res}$, two shoulder peaks appear on both sides of the main peak at $f_{res}$ after Fourier transform. The frequency shifts from the main peak are $\pm f_{tbm}$, respectively, and the peak height is proportional to the measured magnetic field. After magnetic field calibration, the magnitude of the measured low-frequency weak magnetic field signal can be obtained by measuring the height of the shoulder peaks [31,32,33,34]. In general, the limit of detection (LOD) depends on the dimension of the ME sensor and the operating frequency, and the LOD under an AC field is smaller than that under a DC one. For example, the LOD under an AC magnetic field of a metal glass/lead niobium magnesium-Lead titanate (Metglas/PMN-PT) sensor, reported by S. Zhang at University of Wollongong, is 2 pT, while that under a DC magnetic field is only 200 pT [35]. Z. Chu at Peking University also reported an LOD of 115 pT for a Metglas/PMN-PT sensor at 10 Hz [36].

Regarding thin-film ME sensors, they are promising because of their lower operating frequency and magnetic field sensitivity. E. Quandt at Kiel University thoroughly studied film ME sensors composed of soft magnetic films and an AlN ferroelectric layer. He reported that the magnetoelectric properties of AlN/FeCoSiB can be regulated by adjusting the NiTi substrate, and very low frequency of 273 Hz and an LOD of 110 $\mathrm{pT}/\sqrt{\mathrm{Hz}}$ were obtained. In addition, ultrahigh sensitivity of 64 kV/T under a zero bias magnetic field was obtained by means of the converse ME modulation of thin-film Si cantilever composites, and the LODs of 210 $\mathrm{pT}/\sqrt{\mathrm{Hz}}$ at 1 Hz and 70 $\mathrm{pT}/\sqrt{\mathrm{Hz}}$ at 10 Hz were achieved. Even for similar systems, linear resolution of 1.2 nT at 200 mHz can be obtained by combining mesoscale thin-film cantilever beams with pickup coil readout [19,28,37,38]. However, thin-film ME sensors require complex micromachining techniques and have a high cost. The preparation of bulk ME sensors is simple; the technology is flexible, and it has good development prospects. Therefore, this study focused on exploring the bulk ME sensor and its measurement method.

## 2. Experimental Methods

An FeCoSiB metglas sheet with dimensions of 22^length^ × 2^width^ × 0.03^thickness^ mm^3^ was bonded to a lead zirconate titanate Pb(Zr,Ti)O_3_-5B (PZT-5B) sheet with dimensions of 25 × 2 × 0.5 mm^3^ using AB epoxy adhesive produced by 3M company. Mixing AB glue was coated between the Metglas ribbon and the PZT-5B substrate and then placed in a vacuum–heat sealing bag for degassing and sealing using a vacuum–heat sealing machine. The composite was sandwiched between two slides, and they were tightly clamped with plastic clips. The composite was cured at room temperature for 36 h. The PZT-5B sheet was 3 mm longer than the Metglas sheet, because the reserved part of PZT-5B was used to stick the enameled wire to measure the output ME voltage. The upper and lower sides of PZT were coated with silver glue in advance; then, the piezoelectric PZT-5B with piezoelectric coefficient $d_{33}=550\;\mathrm{pC}/N$ was polarized in the thickness direction under a voltage of 3.0 kV.

The assessment of the performance of the ME sensor was carried out using a home-made ME test system, as shown in Figure 1. The procedure is as follows: The prepared ME sensor is put in a long–straight solenoid magnetic field generator. An AC voltage signal provided by a lock-in amplifier (Stanford; model SR830) is amplified by a power amplifier and then input into the above long–straight solenoid to generate AC magnetic field $H_{ac}$. The long–straight solenoid is placed along the axis center of the Helmholtz coils. DC bias magnetic field $H_{bias}$ is generated using Helmholtz coils supplied by DC power supply (YL2420). The variation in $H_{ac}$ (also called the magnetic field to be measured, $H_{tbm}$) causes a strain in the piezomagnetic metglas (as-quenched FeCoSiB amorphous ribbon in this study) due to the magnetostrictive effect. This strain is exerted on the piezoelectric PZT-5B through the AB epoxy adhesive, leading to voltage via the piezoelectric effect, mediated by interface stress. The output voltage of the ME sensor is amplified by the lock-in amplifier and then transmitted and processed using a home-made Labview program. This measurement instrument can be used to explore some performance aspects of ME sensors, for example, to find out the optimal working points, including $f_{res}$ and DC bias field $H_{bias}$, and to measure the LOD, magnetic field resolution, and sensitivity of the ME sensors. In addition, the magnetic field resolution under DC and AC fields at $f_{res}$ or ultra-low frequencies of ME sensors can be measured by improving the instrument. Under a certain DC field (around the optimal DC bias field), the output voltage $V_{out}$-versus-frequency [$V_{out}\left(f\right)$] curve can be obtained by scanning the frequency from 0 to 100 kHz. The frequency at the peak of the $V_{out}\left(f\right)$ curve is $f_{res}$. After that, by setting the lock-in amplifier at $f_{res}$ and scanning DC field H, the $V_{out}\left(H\right)$ curve can be obtained. The magnetic field at maximum $V_{out}$ is the optimal magnetic field, i.e., the optimal $H_{bias}$.

### 2.1. AC Field Resolution at $f_{res}$

The frequency of the lock-in amplifier is set at $f_{res}$. AC magnetic field strength $H_{ac}$, exerted on the ME sensor, can be tuned by the output voltage of the lock-in amplifier. For a fixed $H_{ac}$, a stable ME output voltage $V_{out}$ can be obtained from the lock-in amplifier. Thus, a $V_{out}$ platform appears in the $V_{out}$–time curve. Since $V_{out}$ is proportional to $H_{ac}$, decreasing H gives rise to a series of steps in the $V_{out}$–time curve. The AC field resolution limit can be approached by decreasing $H_{ac}$ till the steps are indistinguishable.

### 2.2. AC Field Resolution at Ultralow-Frequency 

During the measurement of AC field resolution at ultralow frequency, the AC magnetic field power supply of the lock-in amplifier is replaced by a signal generator (RIGOL DG2102), because the signal generator can more easily adjust the signal modulation frequency and depth. However, $V_{out}$ of the ME sensor is still received and processed by the SR830 lock-in amplifier. Although a high AC magnetic field resolution can be achieved at $f_{res}$, it dramatically reduces when the operating frequency is deviated from $f_{res}$. This can be attributed to the rapid reduction in ME coupling efficiency. This characteristic of ME sensors shows that they are not suitable for the direct measurement of ultralow-frequency magnetic fields. Therefore, it is necessary to find out a way to measure ultralow-frequency magnetic fields by taking advantage of the extremely high sensitivity of ME sensors. In this study, a frequency up-conversion technique (FUCT) is proposed for measuring ultralow-frequency magnetic fields using ME sensors. A signal generator outputs a reference signal at $f_{res}$, and another one generates weak magnetic field to be measured $H_{tbm}^{AC}$ at very low frequency ($f_{ac}$ = 8 Hz for instance) via a solenoid. Two signals are fed into long–straight solenoids individually, which in turn apply the different magnetic fields to the ME sensor. The output voltage from the ME sensor is picked up by an oscilloscope (Tektronix MDO34), instead of a lock-in amplifier. It can be seen that two voltage shoulder peaks appear on the left ($f_{res}$ − $f_{ac}$) and right ($f_{res}$ + $f_{ac}$) of $f_{res}$, respectively, with a frequency shift Δf of $\pm f_{ac}$ from $f_{res}$. The voltage amplitude of the shoulder peak is proportional to the strength of low-frequency modulation field $H_{tbm}^{AC}$. AC field resolution $H_{rsl}^{AC}$ can be obtained by reducing the low-frequency $H_{tbm}^{AC}$ till the shoulder peaks are undetectable. By changing the modulation frequency and repeating the above operation, the frequency dependence of $H_{rsl}^{AC}$ can be obtained.

## 3. Results and Discussion

The AC field frequency dependence of $V_{out}$ of the ME sensor was detected by scanning the AC field frequency in the range of 1–70 kHz under a zero DC bias field (Figure 2a). The center frequency of 60.041 kHz was observed. However, when the AC field frequency was scanned under a DC bias field of 2.5 Oe, the peak frequency shifted from 60.041 kHz to 60.101 kHz (Figure 2b). The resonance peak shift can be attributed to the influence of the magnetic field on Young’s modulus, i.e., the delta-E effect [39,40,41]. When comparing Figure 2a,b, it is interesting to note that $V_{out}$ at $H_{bias}$ = 2.5 Oe was dramatically enhanced to twice that under the zero field.

In order to find the optimal DC bias magnetic field, the DC field was scanned in the range of 0–15 Oe at $f_{res}$ of 60.101 kHz. Figure 3 shows the $V_{out}(H_{dc})$ curve and its differential. As illustrated, $V_{out}$ reached the maximum around 2.5 Oe. Therefore, 2.5 Oe was the optimal DC bias field for AC magnetic field measurement ($H_{bias}^{AC}$). The differential ($dV_{out}/dH_{dc}$) curve of the $V_{out}(H_{dc})$ relationship represents the variation rate of $V_{out}$ under the external DC magnetic field. The magnetic field at the maximum of the differential curve ${\left(dV_{out}/dH_{dc}\right)}_{max}$ is the optimal bias field for the DC magnetic field measurement ($H_{bias}^{DC}$), where the output voltage is the most sensitive to the variation in the DC magnetic field. From Figure 3, it can be seen that ${\left(dV_{out}/dH_{dc}\right)}_{max}$ occurred at 0.5 Oe (i.e., $H_{bias}^{DC}$ = 0.5 Oe).

$f_{res}$ of 60.101 kHz and the bias field of 2.5 Oe were adopted as the working points for AC field resolution measurement. The $V_{out}$(time) steps under various input AC fields are shown in Figure 4. The absolute height of the $V_{out}$(time) step refers to output voltage $V_{out}$ at a certain AC voltage input to the long–straight solenoid, which can be converted to the input AC magnetic field intensity $H_{in}$ after calibration. Thus, the relative height of the neighboring $V_{out}$(time) steps can be converted as the difference in the AC field, $\Delta H_{in}$. In other words, if $H_{in}$ is continuously reduced till the step height is indistinguishable, the corresponding $\Delta H_{in}$ is the resolution limit of AC magnetic field $H_{rsl}^{AC}$. As illustrated in Figure 4, an excellent $H_{rsl}^{AC}$ of 0.2 nT was obtained at $f_{res}$ of 60.101 kHz.

Similarly, DC field resolution $H_{rsl}^{DC}$ can be obtained using a DC bias field perturbation (DBFP) method considering the working points of $f_{res}$ = 60.101 kHz and $H_{bias}^{DC}$ = 0.5 Oe [8], as shown in Figure 5. The red dotted line represents output voltage $V_{out}$ of the ME sensor operating at the above working points without the DC perturbation magnetic field $({\left(V_{out}\right)}_{0}$), which can be used as a reference voltage without input DC magnetic field. The optimal DC bias field $H_{bias}^{DC}$ = 0.5 Oe was supplied by the Helmholtz coils. The DC field to be measured, $H_{tbm}^{DC}$, which has an opposite direction with respect to $H_{bias}^{DC}$, was generated by the small solenoid, supplied by a Keithley 2400 source meter. If $H_{tbm}^{DC}$ (for example, 5.0 nT) was exerted on the ME sensor, the $V_{out}$(time) step decreased from 5.97 mV to 2.91 mV (Figure 5). If DC input field $H_{tbm}^{DC}$ was removed, $V_{out}$ was recovered to its original value. By gradually reducing the $H_{tbm}^{DC}$ value till the $V_{out}$(time) step cannot be distinguished from the base line, DC magnetic field resolution $H_{rsl}^{DC}$ can be obtained. As illustrated in Figure 5, a very low $H_{rsl}^{DC}$ of 0.34 nT was achieved in this study.

As discussed above, the optimal AC field sensitivity of the ME sensor occurred at the working points of $f_{res}$ = 60.041 kHz @$H_{bias}^{AC}=$ 2.5 Oe. In order to measure the ultralow-frequency field, the FUCT method is proposed. In this method, another magnetic field, $H_{tbm}^{AC}$, with low frequency, $f_{tbm}$, as a field to be measured is mixed with AC exciting field $H_{ac}$ at resonance frequency, leading to two shoulder peaks located on the left and right sides of the main peak with frequency shift of $\pm f_{tbm}$. The intensity of $H_{tbm}^{AC}$ and frequency $f_{tbm}$ can be tuned by varying the modulation depth (MD = $H_{tbm}^{AC}/H_{ac}$) and the frequency of the signal source [31]. The time base of the oscilloscope sampling is 40 ms, and the recording length is 100 k. Figure 6a,b show the influence of the modulation frequency on the location of the shoulder peaks at an MD of 0.5%. As illustrated in Figure 6a, although the operating frequency is higher than 60 kHz, the modulated shoulder peaks at frequency differences of 0.4, 0.6, and 0.8 Hz are clearly distinguishable. With the further decrease in the modulation frequency, a modulation signal at the ultralow frequency of 50 mHz can also be distinguished from the background signal (Figure 6b). This indicates that the measurement instrument can detect the ultralow-frequency magnetic field at a frequency as low as 50 mHz. By fixing the modulation frequency (1 Hz, for instance), the peak voltage of the shoulder peaks is proportional to the MD (i.e., $H_{tbm}^{AC}$). AC field resolution $H_{rsl}^{AC}$ can be approached by reducing the MD at the modulation frequency till the shoulder peaks cannot be distinguished from the background signal. Using this method, AC field resolutions $H_{rsl}^{AC}$ in the frequency range from 80 Hz to 50 mHz were measured and are summarized in Figure 7. As illustrated, $H_{rsl}^{AC}$ dramatically deteriorated with the decrease in frequency. $H_{rsl}^{AC}$ rapidly increased from 0.45 nT @80 Hz to 51 nT @50 mHz. Nevertheless, at frequencies of over 8 Hz, $H_{rsl}^{AC}$ of the ME sensor was better than 1 nT, which is an excellent achievement [5,38,42,43].

## 4. Conclusions

An FeCoSiB Metglas/PZT-5B magnetoelectric coupling sensor with dimensions of 25 × 2 × 0.5 mm^3^ and resonance frequency of 60.101 kHz was fabricated for weak magnetic field measurement. It is interesting that excellent AC and DC magnetic field resolutions as low as 0.2 and 0.34 nT, respectively, were obtained. The frequency up-conversion technique is an effective way to sensitively measure ultralow-frequency AC magnetic field resolution. It was revealed that the AC field resolution rapidly deteriorated with the decrease in frequency due to the far deviation from the resonance frequency. Nevertheless, excellent AC field resolution was achieved, and the frequency of magnetic field resolution better than 1 nT consistently persisted up to 8 Hz.

## Figures and Tables

**Figure 1 sensors-23-01702-f001:**
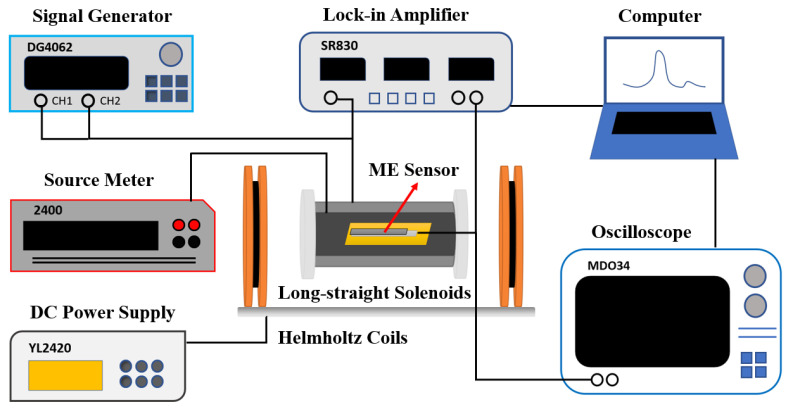
Schematic showing the principle of the ME test system (the actual wiring method can be adjusted according to the test types).

**Figure 2 sensors-23-01702-f002:**
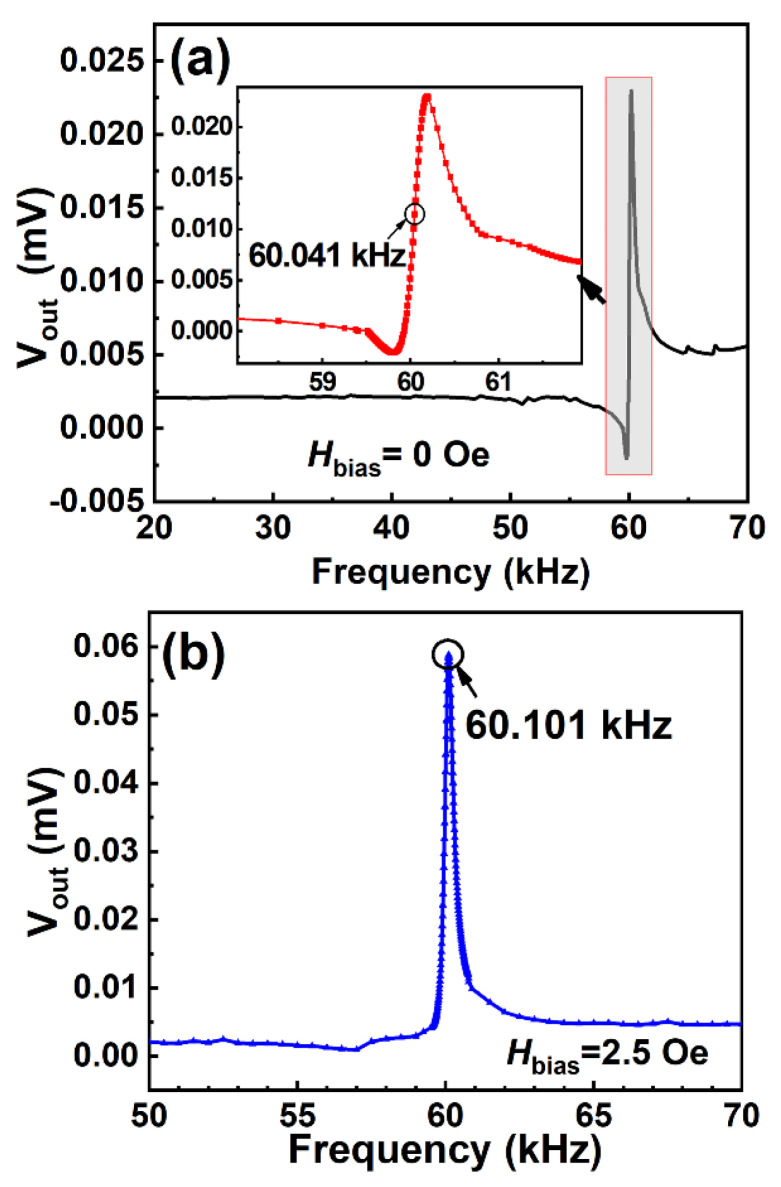
The frequency dependence of $V_{out}$ of the ME sensor without DC bias field $H_{bia}$ = 0 Oe (**a**) and with $H_{bias}$ = 2.5 Oe (**b**). The inset of Figure 2a is a magnified view of the vicinity of the resonance peak.

**Figure 3 sensors-23-01702-f003:**
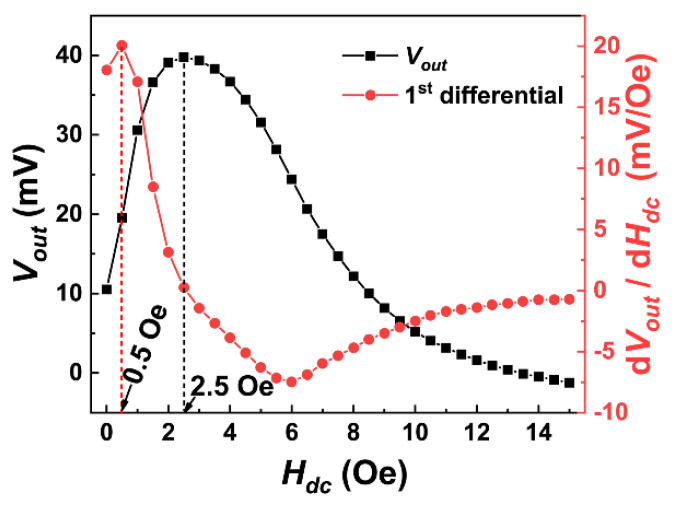
The $V_{out}(H_{dc})$ curve and its differential curve at $f_{res}$ of 60.101 kHz.

**Figure 4 sensors-23-01702-f004:**
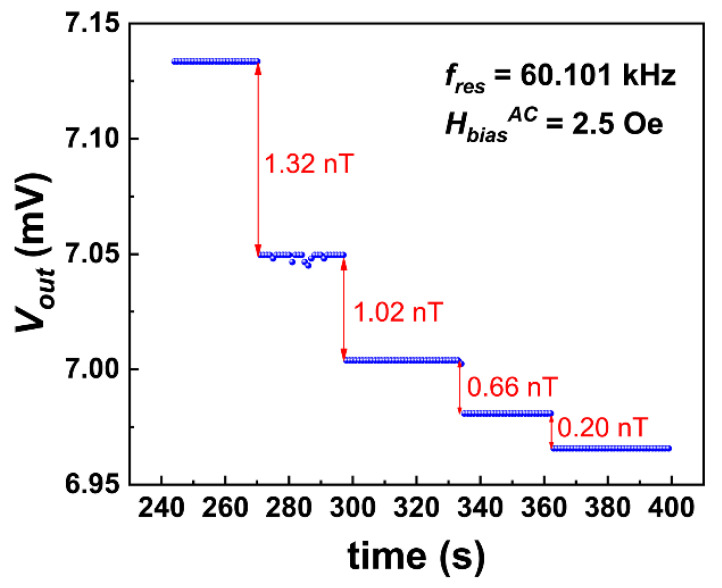
AC magnetic field resolution at $f_{res}$ of 60.101 kHz and $H_{bias}^{AC}$ of 2.5 Oe.

**Figure 5 sensors-23-01702-f005:**
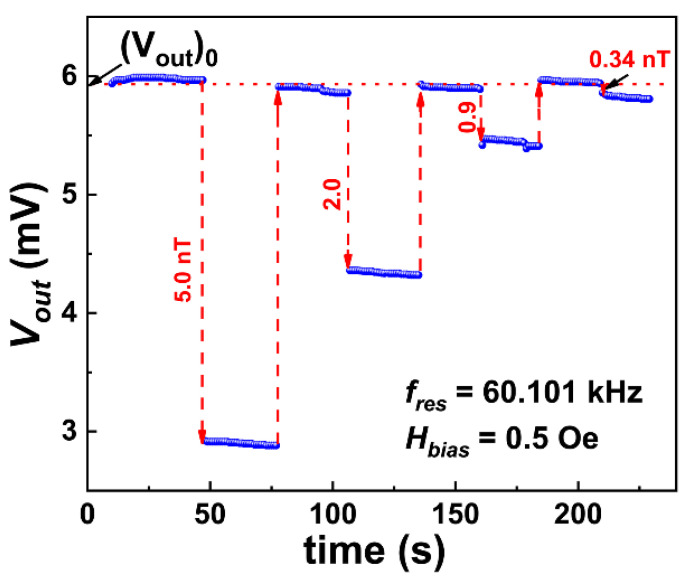
DC magnetic field resolution $H_{rsl}^{DC}$ at $f_{res}$ = 60.101 kHz and $H_{bias}^{DC}$ = 0.5 Oe.

**Figure 6 sensors-23-01702-f006:**
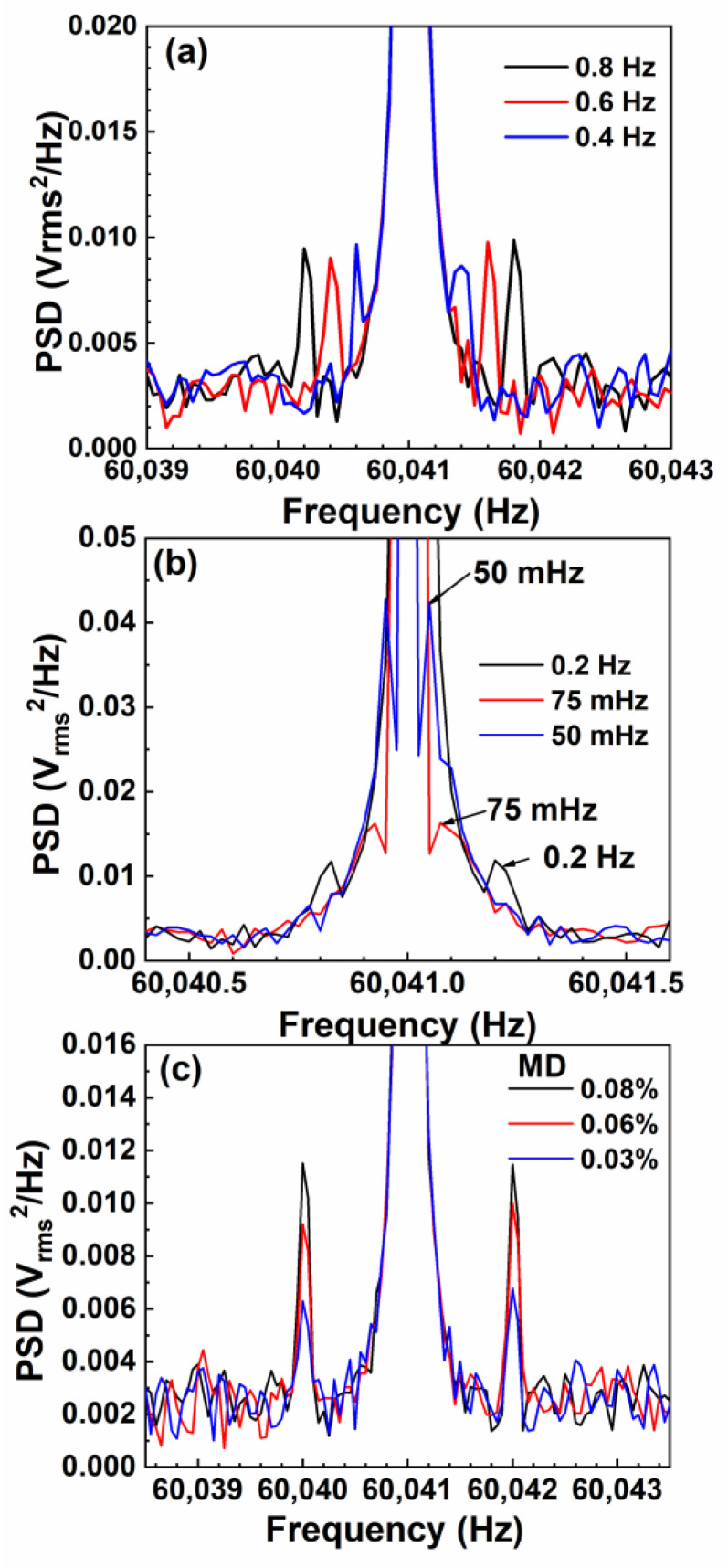
Modulation peaks at MD = 0.05% at various modulation frequencies (**a**,**b**) and at different MDs at a modulation frequency of 1 Hz (**c**).

**Figure 7 sensors-23-01702-f007:**
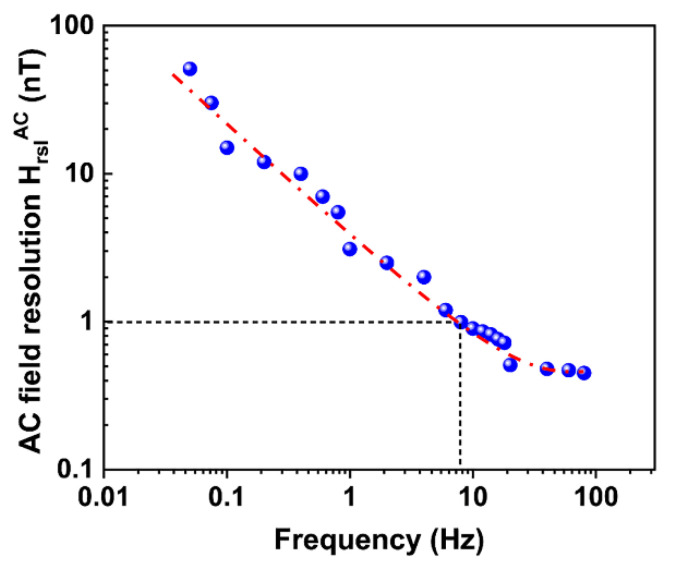
AC magnetic field resolution at ultralow frequencies (the dash-dot line is a guide for the eyes).

## Data Availability

The data presented in this study are not publicly available at this time, but may be obtained upon reasonable request from the authors.

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
