# Peer review of "High-Resolution Magnetoelectric Sensor and Low-Frequency Measurement Using Frequency Up-Conversion Technique"

_sensors, 2023, doi:10.3390/s23031702_

Round 1
Reviewer 1 Report
The authors presented a FeCoSiB metglas/PZT-5B bulk magnetoelectric (ME) sensor and demonstrated its ability for measuring low-frequency magnetic fields. While analyzing the amplitude change of the sensor output at its resonance frequency as parameter to identify the detection limit, the authors claimed sub-nT sensitivity for both AC and DC modes. For low frequency measurement, the authors further characterized the magnitudes of the sidebands around the resonance frequency and highlighted frequency-dependent sensitivity of the ME sensor. Here, I noted few points which should be resolved for improving readability of the manuscript, as follow:
1. “The high resolution ME sensor with sub-pT level is…”, what is the authors’ intension to put the sentence on the abstract? There is no evidence that the evaluated ME sensor is capable of measuring femtoTesla magnetic fields.
2. “The magnetoresistance-type sensors have relatively lower magnetic field resolution”, this statement is incorrect. MR sensors such TMR sensor can even reach remarkable sub-pT sensitivity. See M. Oogane et al 2021 Appl. Phys. Express 14 123002.
3. There should be no different between mechanical resonance frequency fres (Line 47) and resonant frequency fre (Line 60), thus redefinition is unnecessary.
4. Not to confuse Vout-f (Line 115) as mathematical expression, it should be written as Vout(f). Please revise other similar expressions throughout the manuscript, as well as for Vout-H and Vout-Hdc.
5. “Set the frequency of lock-in amplifier…” (Line 120), avoid using instruction as the readers have no obligation to repeat the authors` work.
6. Correct typo for “caliberation” (Line 194).
7. For Fourier transform of the signals, either sampling rate or record length of the oscilloscope defines the resolution of the spectrogram. Please specify those parameters when discussing Fig. 6.
Author Response
The authors presented a FeCoSiB metglas/PZT-5B bulk magnetoelectric (ME) sensor and demonstrated its ability for measuring low-frequency magnetic fields. While analyzing the amplitude change of the sensor output at its resonance frequency as parameter to identify the detection limit, the authors claimed sub-nT sensitivity for both AC and DC modes. For low frequency measurement, the authors further characterized the magnitudes of the sidebands around the resonance frequency and highlighted frequency-dependent sensitivity of the ME sensor. Here, I noted few points which should be resolved for improving readability of the manuscript, as follow:
Q1. “The high resolution ME sensor with sub-pT level is…”, what is the authors’ intension to put the sentence on the abstract? There is no evidence that the evaluated ME sensor is capable of measuring femtoTesla magnetic fields.
Answer: Thank you very much for your reminding. We expressed it wrong here, the measurement accuracy of the studied ME sensor is several hundreds of pT. The correct expression would be sub-nT level. It has been corrected in Line 17 in the revision.
Q2. “The magnetoresistance-type sensors have relatively lower magnetic field resolution”, this statement is incorrect. MR sensors such TMR sensor can even reach remarkable sub-pT sensitivity. See M. Oogane et al 2021 Appl. Phys. Express 14 123002.
Answer: Thank you for recommending such a good reference. It is true that traditional magnetoresistance sensors, such as Hall, AMR, have relatively lower sensitivity, but the sensitivity of TMR sensors has been greatly improved in recent years. In the revised version, the corresponding sentence is changed to “Most of magnetoresistance sensors have relatively lower magnetic field sensitivity, except recently the sensitivity of tunneling magnetoresistance sensors has been greatly improved to the order of sub-pT [7]” (Lines 33-35).
Q3. There should be no different between mechanical resonance frequency fres (Line 47) and resonant frequency fres (Line 60), thus redefinition is unnecessary.
Answer: You're right. After the first definition of the mechanical resonance frequency , all the following are replaced by , and the mechanical resonance frequency no longer appears.
Q4. Not to confuse Vout-f (Line 115) as mathematical expression, it should be written as Vout(f). Please revise other similar expressions throughout the manuscript, as well as for Vout-H and Vout-Hdc.
Answer: OK. All mathematical expressions are uniformly in the format you suggest.
Q5. “Set the frequency of lock-in amplifier…” (Line 120), avoid using instruction as the readers have no obligation to repeat the authors` work.
Answer: Thank you for your correction. The sentence has been corrected as “The frequency of lock-in amplifier is set at fres” (Line 126).
Q6. Correct typo for “caliberation” (Line 194).
Answer: Thanks a lot. Corrected as “calibration” (Line 189 in revision).
Q7. For Fourier transform of the signals, either sampling rate or record length of the oscilloscope defines the resolution of the spectrogram. Please specify those parameters when discussing Fig. 6.
Answer: Good comment. In this study, “The time base of oscilloscope sampling is 40 ms, and the recording length is 100 k.” This discussion has been appended in Lines 218-219 in revision.

Reviewer 2 Report
Paper is devoted studies of metglas/PZT composite using low-frequency using frequency up-conversion technique for ME measurements. The method and experiment design are well desribed.
Reviewer recommends to upgrade ref list and add some details in composite fabrication. How the metglass and PZT were glued. Did author use a clamping in during fabrication. What was the temperature and drying time of the composite?
Author Response
Paper is devoted studies of metglas/PZT composite using low-frequency using frequency up-conversion technique for ME measurements. The method and experiment design are well described.
Reviewer recommends to upgrade ref list and add some details in composite fabrication. How the metglass and PZT were glued. Did author use a clamping in during fabrication? What was the temperature and drying time of the composite?
Answer: The authors thank you for your positive evaluation. The preparation procedure of ME sensor can be described as follows: “The mixing AB glue is coated between Metglas ribbon and PZT-5B substrate, and then placed in a vacuum heat seal bag for degassing and sealing using a vacuum heat seal machine. The ME composite was sandwiched between two glass slides and then placed between pressure anvils to cure at room temperature for 36 h.” The description has been appended in Lines 94-98 in revision.

Reviewer 3 Report
Dear Authors and Editors,
First, there are some big flaws in this manuscript, please improve this manuscript and answer my following questions. Second, the experiments in section 2.2 need to be revised or removed before this manuscript can be published, because this section may mislead the readers. Finally, when we talk about the sensing performance or the lowest detection limit, we need to investigate or discuss the intrinsic NOISE of the sensors. I encourage the authors to read more papers about the noise in magneto-electric sensors from the references and the references of references. It helps.
Q1: About the title. The proposed magnetic sensor is "magneto(elasto)electric" or “magneto-electric”, instead of “magnetoelectric”.
Q2: Line 10 in the abstract. The magneto-electric (ME) sensors include the passive mode and active (frequency up-conversion or nonlinear modulation or H field modulation) mode. Only the passive mode ME sensors have low power consumption. The proposed mode is active mode ME sensors that have a similar power consumption as Fluxgates.
Q3: Line 31. Giant Magneto-Impedance (GMI) and ME sensors are also flux-type magnetometers, as I know.
Q4: Line 37. I invite the authors to read more papers that have been published since 2011. The detection performance (we also call it the equivalent magnetic noise or the equivalent input magnetic noise spectral density, EMN) of ME sensors do depend on the volume of the sensor heads. The performance is directly proportional to the root volume of the sensor head.
Q5: Line 40. The geomagnetic field detection and brain magnetic field imaging usually require an EMN level below 1pT/root-Hz @ 1Hz. The best passive ME sensors have an EMN of several pT/root-Hz@1Hz, and the best active mode ME sensors (the proposed frequency up-conversion technique in this manuscript) have a noise level of several 10pT/root-Hz@1Hz. Both modes need to be improved further again to satisfy the noise requirement of the mentioned applications.
Q6: Line 48. In this manuscript, the proposed ME sensor with frequency up-conversion is an active mode ME sensor. Unlike the passive mode sensors that are listed in the refs, the traditional optimal DC bias for the active mode has been already reported as a value near ZERO DC field.
Q7: Line 51. Quandt et al. studied the thin-film ME composites, however, the proposed ME sensor in this manuscript is a laminated (bulk) ME sensor based on the FeCoSiB metallic glasses and PZT ceramics. There are already many published papers about the active mode ME sensors with metglas and PZT. So, it is suggested to compare the performance with those metglas/PZT laminate ones, rather than the FeCoSiB/AlN thin-films.
Q8: Line 68. The best detection performance of bulk (laminate) ME sensors are better than thin-film ME sensors for either AC or DC signals. Please find mode references. Equivalent magnetic noise levels of 10f-100fT/root-Hz@resonance and several 10pT/root-Hz@1Hz have been achieved since more than ten years ago. Please read more references.
Q9: Line 90. Please detail the manufacturer of FeCoSiB. The FeCoSiB is as-quenched ones or annealed ones? The magnetomechanical properties of FeSiB ribbons is higher than FeCoSiB ones, please explain why did the authors choose the FeCoSiB ones.
Q10: Line 95. Please detail the piezoelectric constant (dp) or the piezoelectric coupling factor (kp) for the PZT5 after the electric poling.
Q11: Line 107. In this manuscript, the output voltage from the piezoelectricity is directly connected to the lock-in amplifier, so the intrinsic noise level of SR830 need to be discussed. Besides, how to make sure that the noise level in the lock-in amplifier is smaller than the noise level in ME senor? Actually, a low-noise voltage or charge pre-amplifier is needed to measure the LOD in a ME sensor. Most of the investigators in the field of high sensitivity magnetic sensors did so.
Q12: Line 152. To supply the mixed modulation signals of low frequency and resonance frequency to the ME laminate is totally wrong. The correct experiment is to feed an individual low-frequency signal (for example 8Hz) and an individual excitation signal at the resonant frequency, as such, a mixed signal can be produced at the output side of the ME sensor by mean of the nonlinearity in the ME composite. The signal mixture occurs in the ME laminate, in another word, if the mixed signals is injected into the sensor head, the experiments make no sense at all. This section MUST be revised or removed before the manuscript can be published.
Q13: Line 162. In Fig.1, an input resistor with a large value, for example 10kOhm or 100kOhm is needed to be connected in series with the input coils, if the input source is a voltage one.
Q14: Line 189. In Fig. 3, the 0.5 Oe shift from the zero point might be due to the residual stress on the FeCoSiB from the PZT part during the epoxy curing. It is suggested that more foils (for example 2 or 3 foils) epoxied together can avoid this shift on the 1st differential curve. Besides, there is an optimal ratio between the thickness of the magnetostrictive/piezoelectric layers. A 30-micro-meter thick ribbon CANNOT match to the thickness of 0.5mm for PZT5 to optimize the ME coupling coefficient. There exists an optimal thickness ratio for every ME laminate sample, considering the chosen magnetostrictive and piezoelectric materials.
Author Response
First, there are some big flaws in this manuscript, please improve this manuscript and answer my following questions. Second, the experiments in section 2.2 need to be revised or removed before this manuscript can be published, because this section may mislead the readers. Finally, when we talk about the sensing performance or the lowest detection limit, we need to investigate or discuss the intrinsic NOISE of the sensors. I encourage the authors to read more papers about the noise in magneto-electric sensors from the references and the references of references. It helps.
Answer: Thank you very much for your valuable comments, which are improving the manuscript a lot.
Experiments in section 2.2 has been removed to avoid misleading readers since the detailed description has been reported in our previous work (Ref. [8]). This work is cited as a reference.
We read some references about the discussion of the sensor noise, studied them carefully. They are beneficial for the modification the manuscript.
Q1: About the title. The proposed magnetic sensor is "magneto(elasto)electric" or “magneto-electric”, instead of “magnetoelectric”.
Answer: Thanks. We use the word “magneto-electric” as your suggestion. All the “magnetoelectric” have been changed as “magneto-electric” in revision.
Q2: Line 10 in the abstract. The magneto-electric (ME) sensors include the passive mode and active (frequency up-conversion or nonlinear modulation or H field modulation) mode. Only the passive mode ME sensors have low power consumption. The proposed mode is active mode ME sensors that have a similar power consumption as Fluxgates.
Answer: You are right. The proposed ME sensors in this study are working in an active mode with a relatively high power consumption. To avoid confusion, the words “…, low power consumption,…” in the first sentence in abstract “…with ultrahigh sensitivity, low power consumption, and …” were deleted.
Q3: Line 31. Giant Magneto-Impedance (GMI) and ME sensors are also flux-type magnetometers, as I know.
Answer: As for the classification of magnetic field sensors, we also refer to the classification method from literatures. For the flux-type sensors, such as SQUID, it is sensitive to the magnetic flux passing through the superconducting loop, while the magnetic field type magnetic sensor is sensitive to the strength of the magnetic field surrounding the sensor, rather than the magnetic flux. It is believed that ME sensor is derived from magnetostrictive effect, and its sensitive physical parameter is magnetic field rather than magnetic flux. Of course, magnetic field and magnetic flux are generally positively correlated, rather than completely independent.
Q4: Line 37. I invite the authors to read more papers that have been published since 2011. The detection performance (we also call it the equivalent magnetic noise or the equivalent input magnetic noise spectral density, EMN) of ME sensors do depend on the volume of the sensor heads. The performance is directly proportional to the root volume of the sensor head.
Answer: As you said, the performance of ME sensor is also related to the sensor dimension. The smaller the sensor is, the lower the sensitivity will be. However, compared with the flux-type sensors, the dimension dependence is not so strong, which is beneficial to its miniaturization. This sentence is changed as “Although the performance of the ME sensor also depends on its size, its size dependence is not as strong as that of the flux-type sensor, which is beneficial to its miniaturization.” (Lines 38-40 in revision).
Q5: Line 40. The geomagnetic field detection and brain magnetic field imaging usually require an EMN level below 1pT/root-Hz @ 1Hz. The best passive ME sensors have an EMN of several pT/root-Hz@1Hz, and the best active mode ME sensors (the proposed frequency up-conversion technique in this manuscript) have a noise level of several 10pT/root-Hz@1Hz. Both modes need to be improved further again to satisfy the noise requirement of the mentioned applications.
Answer: It is true that the sensitivity of ME sensor cannot meet the requirements of geomagnetic field and magnetoencephography (MEG) measurement at present, but in recent years, the rapid development of ME sensor research makes it possible that in the near future ME sensor will be able to meet the needs of geomagnetic field and MEG. The corresponding sentence is changed as “At present, ME sensors have been used for non-contact positioning and navigation [14-20]. With the continuous progress of technology, ME sensor is expected to be applied in the geomagnetic field and magnetoencephography in the near future” (Lines 40-43 in revision).
Q6: Line 48. In this manuscript, the proposed ME sensor with frequency up-conversion is an active mode ME sensor. Unlike the passive mode sensors that are listed in the refs, the traditional optimal DC bias for the active mode has been already reported as a value near ZERO DC field.
Answer: Indeed, some researchers, such as V. G. Harris et al. in Northeastern University, have reported that the DC bias field is small or close to zero in active mode. We tried the relationship between the strength of the bias field and the output signal, and learned that it can be got the better performance under appropriate DC bias field for the studied sample. In order to make it clear, we added a sentence “(For active modes, the bias field may be as small as zero)” (Lines 52-53 in the revision).
Q7: Line 51. Quandt et al. studied the thin-film ME composites, however, the proposed ME sensor in this manuscript is a laminated (bulk) ME sensor based on the FeCoSiB metallic glasses and PZT ceramics. There are already many published papers about the active mode ME sensors with metglas and PZT. So, it is suggested to compare the performance with those metglas/PZT laminate ones, rather than the FeCoSiB/AlN thin-films.
Answer: Lines 50-54 are to state that current research work of ME sensors is mainly focused on reducing and improving . Quandt's work was cited as an example because of its and lower less than 1 kHz. For the sake of comprehensiveness, we also add an example of a laminated structure sample with a high . (Lines 53-58)
Q8: Line 68. The best detection performance of bulk (laminate) ME sensors are better than thin-film ME sensors for either AC or DC signals. Please find mode references. Equivalent magnetic noise levels of 10f-100fT/root-Hz@resonance and several 10pT/root-Hz@1Hz have been achieved since more than ten years ago. Please read more references.
Answer: Thank you for your correction. We read more references and found that “the LOD for the thin-film sensor is better than that of bulk one” is not appropriate. So it is deleted in the revision.
Q9: Line 90. Please detail the manufacturer of FeCoSiB. The FeCoSiB is as-quenched ones or annealed ones? The magnetomechanical properties of FeSiB ribbons is higher than FeCoSiB ones, please explain why did the authors choose the FeCoSiB ones.
Answer: The as-quenched FeCoSiB Metglas ribbon used in this study was donated by Qingdao Yunlu Company. The preparation method is the melt spinning. Our laboratory does not have the equipment to prepare Metglas independently, so we can only purchase or get it from a friend.
Q10: Line 95. Please detail the piezoelectric constant (dp) or the piezoelectric coupling factor (kp) for the PZT5 after the electric poling.
Answer: The piezoelectric coupling factor of PZT-5B substrate is d33=550 pC/N, which is appended in the revision. (Lines 101-102).
Q11: Line 107. In this manuscript, the output voltage from the piezoelectricity is directly connected to the lock-in amplifier, so the intrinsic noise level of SR830 need to be discussed. Besides, how to make sure that the noise level in the lock-in amplifier is smaller than the noise level in ME senor? Actually, a low-noise voltage or charge pre-amplifier is needed to measure the LOD in a ME sensor. Most of the investigators in the field of high sensitivity magnetic sensors did so.
Answer: In this study, the noise of SR830 used in ME testing is sufficiently small compared with the sensor signal. In addition, the noise of SR830 and other noises have been reasonably corrected according to the normal calibration conditions when compiling the Labview related programs.
Q12: Line 152. To supply the mixed modulation signals of low frequency and resonance frequency to the ME laminate is totally wrong. The correct experiment is to feed an individual low-frequency signal (for example 8Hz) and an individual excitation signal at the resonant frequency, as such, a mixed signal can be produced at the output side of the ME sensor by mean of the nonlinearity in the ME composite. The signal mixture occurs in the ME laminate, in another word, if the mixed signals is injected into the sensor head, the experiments make no sense at all. This section MUST be revised or removed before the manuscript can be published.
Answer: I agree with you. At the beginning of the study, we did it according to the description of the paper. Later, we realized that the measurement was wrong, so we separated the reference signal and the measured signal, input them separately to the ME sensor, and coupled them in the ME sensor, so that we could get the correct signal. Sorry, I made a mistake when I wrote the manuscript. We have corrected this part in revision to " A signal generator outputs a reference signal at , and another one generates weak magnetic field to be measured with very low frequency ( =8 Hz for instance) via a solenoid. Two signals are fed into long straight solenoids individually, which in turn applies the different magnetic fields to the ME sensor." (Lines 145-148).
Q13: Line 162. In Fig.1, an input resistor with a large value, for example 10kOhm or 100kOhm is needed to be connected in series with the input coils, if the input source is a voltage one.
Answer: In this study, the coil does not have a large resistance in series, because the signal output power is very small. If the resistance is increased, the current is too small, resulting in a weak magnetic field.
Q14: Line 189. In Fig. 3, the 0.5 Oe shift from the zero point might be due to the residual stress on the FeCoSiB from the PZT part during the epoxy curing. It is suggested that more foils (for example 2 or 3 foils) epoxied together can avoid this shift on the 1st differential curve. Besides, there is an optimal ratio between the thickness of the magnetostrictive/piezoelectric layers. A 30-micro-meter thick ribbon CANNOT match to the thickness of 0.5mm for PZT5 to optimize the ME coupling coefficient. There exists an optimal thickness ratio for every ME laminate sample, considering the chosen magnetostrictive and piezoelectric materials.
Answer: Thank you for your valuable comments. The FeCoSiB ribbon with a thickness of 30 μm does not match the PZT-5B substrate with a thickness of 0.5 mm. We are using Comsol software to simulate the thickness and size matching relationships between piezomagnetic and piezoelectric materials. Actually, we have also tried to glue several pieces of FeCoSiB ribbons together to achieve a match with PZT substrate. However, due to uneven surface of FeCoSiB ribbons and/or uneven bonding among the ribbons, the bonding effect is not good as we expected. Anyway, you have given us a good suggestion. In the follow-up study, we will focus on the size matching relationship between piezomagnetic and piezoelectric materials, so as to realize the optimization of magnetoelectric coupling effect and optimal ME performance.

Round 2
Reviewer 3 Report
The Q12 was the catastropic flaw in this manuscrit. However, the authors' respone declared that this problem had been realized and fixed for their experiments. And "they just made a mistake when they drafted this manuscript."
Based on this statement, I believe that the manuscript is able to be published. I leave it to the editor for a final decision.